# Intradermal Testing Doubles Identification of Allergy among 110 Immunotherapy-Responsive Patients with Eustachian Tube Dysfunction

**DOI:** 10.3390/diagnostics11050763

**Published:** 2021-04-24

**Authors:** David S. Hurst, Bruce R. Gordon, Alan B. McDaniel, Dennis S. Poe

**Affiliations:** 1Department of Otolaryngology, Tufts University, Boston 02111, MA, USA; 2Department of Laryngology & Otology, Harvard University, Boston, MA 02114, USA; brgordon@capecodhealth.org; 3Department of Otolaryngology, University of Louisville, Louisville, KY 40202, USA; abmcdaniel0621@gmail.com; 4Harvard Medical School, Boston, MA 02114, USA; dennis.poe@childrens.harvard.edu

**Keywords:** eustachian tube dysfunction, otitis media, skin prick test, asthma, allergic rhinitis, allergy testing, allergy immunotherapy, intradermal test

## Abstract

The purpose of this study was to determine whether the sensitivity advantage of intradermal dilutional testing (IDT) is clinically relevant in patients with obstructive Eustachian tube dysfunction (ETD) or otitis media with effusion (OME). This retrospective, private-practice cohort study compared the sensitivity of skin prick tests (SPT) vs. IDT in 110 adults and children with suspected allergy and OME. *Primary outcome* measure was symptom resolution from allergy immunotherapy (AIT). IDT identified 57% more patients as being allergic, and 8.6 times more reactive allergens than would have been diagnosed using only SPT. Patients diagnosed by IDT had the same degree of symptom improvement from immunotherapy, independent of allergen sensitivity (66% by SPT vs. 63% by IDT; *p* = 0.69, not different). Low-sensitivity allergy tests, which may fail to identify allergy in over two thirds of children aged 3 to 15 as being atopic, or among 60% of patients with ETD, may explain why many physicians do not consider allergy as a treatable etiology for their patient’s OME/ETD. IDT offers superior sensitivity over SPT for detecting allergens clinically relevant to treating OME/ETD. These data strongly support increased utilization of intradermal testing and invite additional clinical outcome studies.

## 1. Introduction

Allergy tests detect hypersensitivity to allergens suspected of triggering symptoms, but are all skin tests equivalent in their responses? Skin prick tests (SPT) are often the only tests used to diagnose allergy, because intradermal dilutional tests (IDT) are thought by some to offer no additional relevant information [1]. However, the subsequent decision to treat with allergy immunotherapy (AIT), and the resulting therapeutic response, depend on the ability of the chosen test to identify truly allergic individuals and their significant allergens [2]. Yet, physicians face a conundrum in diagnosing patients with classic signs and symptoms of allergy when SPT are negative. 

This study was designed to determine whether the difference in sensitivity between IDT and SPT is clinically relevant in patients with obstructive Eustachian tube dysfunction (ETD), presenting as barochallenge, tympanic retraction, or OME. We compared the AIT responses of low-sensitivity patients who were identified only by IDT, that is, those who have had negative results with SPT and those showing positive results with IDT; i.e., low-sensitivity (SPT−/IDT+) vs. high-sensitivity patients (SPT+).

### Relationship of Allergy to ETD

Current best evidence supports an association between allergic rhinitis (AR) and ETD. Meta-analysis suggests a strong correlation between AR and OME among children [3]. Yet, not all patients with ETD suffer from AR, and vice versa [4]. Exposure to allergen challenge has consistently resulted in a dose-dependent decrease in ET patency, regardless of whether a seasonal (ragweed) or perennial (dust mite) allergen challenge was used [5]. Evaluation of the records of 2.4 billion pediatric visits found allergy to be associated with a 2- to 4.5-fold increased incidence of OME [6]. Histologic, epidemiologic, and clinical studies based on objective allergy testing have thus far established that (1) the majority of OME patients are atopic [7]; (2) all the mediators necessary for a Th2 allergic response are present in the middle ear [8,9]; (3) per the 2016 guidelines, the middle ear is part of the unified airway, and “like other parts of respiratory mucosa, the mucosa lining the middle-ear cleft is capable of an allergic response” [10]; and, finally, (4) chronic middle ear patients’ disease partially or completely resolves with AIT based on intradermal testing results [11], or from food elimination diets [12]. 

Many OME clinical studies over the past 75 years have shown both an increased association of allergy symptoms (rhinitis, asthma, and eczema) and positive in vivo or in vitro tests for IgE [13]. The reported incidence of allergy being related to ETD and/or OME, as determined by allergy testing gives mixed results, ranges from 15% to 93% in pediatrics and up to 35% among adults [8]. This wide variation in incidence could be due in part to differences in testing methods. 

Some clinicians feel that low-sensitivity allergy detected by IDT is not clinically meaningful, citing concerns that low-sensitivity patients require more concentrated antigen for testing which can result in false positives [1].

The 2016 Clinical Guidelines Update [10] stated that despite “a high prevalence of atopic conditions, such as AR, in children with OME, there are no benefits to routinely treating with antihistamines, decongestants, or steroids (systemic or topical intranasal).” Yet, a more recent systematic review of 3010 papers found that “clinical evidence and analyses of biomarkers suggested that allergy may be linked to some phenotypes of otitis media and, in particular, to otitis media with effusion and acute re-exacerbations in children with middle ear effusion” [14]. 

We believe that the results of many of these prior studies, which are dependent on the sensitivity of the type of allergy test used, have underestimated the true incidence of allergy among OME patients by using tests with poor sensitivity. This study was designed to test the hypothesis: IDT, compared to SPT, has greater sensitivity for detecting allergens that are *clinically relevant* in patients with ETD and/or OME. Our test of hypothesis was by evaluation of each patient’s AIT response, as reflected in his or her perceived symptom improvement. Treatment response was dependent on treatment allergens being correctly chosen by their skin tests.

## 2. Materials and Methods

This is a retrospective study of 110 patients in a solo, community-based practice who presented with chronic (≥3 months) symptoms or signs of obstructive ETD that meet the criteria as defined in a Clinical Consensus Statement [15]. Assessments included history, otologic exam, and pneumatic otoscopy. Symptoms of obstructive ETD included aural fullness, aural pressure, and otalgia, often associated with hearing loss, or OME as documented by audiometry and tympanometry demonstrating evidence of negative middle ear pressure, conductive hearing loss, and/or effusion. Seventy-six of these 110 (69%) had previously been treated with a total of 182 tympanostomy tube placements (TTP) (87% of children, 66% of adults), mean 2.56 per patient. Resolution of OME was also confirmed by both audiometry and tympanometry returning to normal. 

ETD, asthma, and AR were diagnosed based on clinical symptoms, compatible physical findings, and positive skin tests [16]. All patients diagnosed with obstructive ETD and who opted for AIT were included in this study. Ethics approval was obtained from the Franklin Memorial Hospital Committee on Ethics and Human Experimentation (Farmington, ME 04938, USA, Personal letter June, 2006). Informed consent for allergy testing was obtained from the patient or parent for both testing and treatment. 

### 2.1. Allergy Testing

All patients were tested by the primary author using multi-dilution IDT according to current practice parameters [17,18] for *Dermatophagoides pteronyssinus*, *Dermatophagoides farinae*, cat, dog, American cockroach (*Periplaneta Americana*), grass (timothy or meadow fescue), tree (birch or oak), ragweed, goldenrod, lambs quarters, *Alternaria alternata*, and *Cephalosporium acremonium*.

Following the 2008 Updated Practice Parameters, which state that “A suggested way of determining appropriate Intracutaneous test concentrations is a serial end point titration regimen” [19], patients were tested with 4mm intradermal wheals [20] and with glycerin-matched control tests [21]. IDT test allergens were serially diluted five-fold, six times, from standardized extracts or the highest concentration available, usually 1:20 *w*/*v* (Greer, Lenoir, NC 28633, USA). Testing was begun at dilute allergen concentrations with Dilution 5 (D5); 1:62,500 *w*/*v* or Dilution 4 (D4); 1:12,500 *w*/*v*; and continuing up to the most concentrated dilution tested (D2), 1:500 *w*/*v*. Intracutaneous tests of wheal growth were measured 10 to 15 min after injection and both wheal and erythema (in millimeters) were recorded [19,20]. Positive test or “end point” was defined as the lowest concentration of allergen that produces a wheal: (1) that the first wheal is 2 mm larger than the negative control wheal and (2) is followed by a second wheal that is at least 2 mm larger than the preceding one [17,19,20]. Concentration-matched glycerin controls prevented the misinterpretation of skin-wheal responses and reduced false positive results [21]. Patients who were completely negative for all skin tests were not offered immunotherapy and were excluded from the study, as were those with craniofacial abnormalities, muscular dystrophy, history of previous cholesteatoma, autoimmune disorder, or those who only had in-vitro testing. All eligible patients were offered the same treatment options.

Thirty-nine patients had SPT performed by AAAAI board certified physicians within two years prior to enrollment. Records of these skin tests were obtained. These patients were re-tested by IDT and are included in this study.

In-vitro testing had been used to screen for allergic sensitive patients who were later skin tested. Only a very few infants or needle phobic patients were treated from their Pharmacia CAP or RAST results and as such were excluded from this study as those results are difficult to extrapolate to skin test results.

Subgroup A patients reacted to allergens only on the strongest concentration tested, D2 (1:500 *w*/*v*). Subgroup B reacted to one allergen at D3 (1:2500 *w*/*v*) and all others to dilution D2. Subgroup C reacted to more than one allergen at D3 and the rest at D2. Subgroup D reacted to a single allergen at D4 (1:12,500 *w*/*v*), with all other positive allergens identified at stronger concentrations. Subgroup E were the most sensitive, reacting to at least two allergens at dilution D4 or weaker. 

Skin Test Responses: A = all D2 (1:500 *w*/*v*), B = D2 and 1 D3 (1:2500 *w*/*v*), C = D2 and D3, D = at least 1 D4 (1:12,500 *w*/*v*), and E = 2 or more D4. Percent improvement is: (pretreatment symptom score minus symptom score after AIT)/pretreatment symptom score.

### 2.2. SPT Status

Positive IDT dilution results were categorized as being high-sensitivity reactors (SPT+/IDT+) or low-sensitivity reactors (SPT−/IDT+) using the known sensitivity of the Multi-test II (Lincoln Diagnostics, Decatur, IL 62526, USA) as being between IDT D3 (1:2500 *w*/*v*) and D4 (1:12,500 *w*/*v*) [22,23]. These groupings are consistent with the 2008 Updated Practice Parameters, which state “comparative equivalency studies based on history and symptoms alone revealed that IDT is roughly equivalent to new skin prick tests only at dilutions ranging from 1:12,500 (*w*/*v*) to 1:312,000 (*w*/*v*)” [19] (Pg. SS 24). Only IDT when used at 1:500 *w*/*v* can physically introduce enough allergen to reliably detect most low-sensitivity allergies.

SPT results for each antigen tested were inferred from the concentration of each IDT positive endpoint reaction. Therefore, Subgroups D and E were considered SPT positive “high-sensitivity reactors” as they responded to at least one 1:12,500 *w*/*v*, or more dilute allergen solutions, while those in Subgroups A, B, and C were considered to be “low-sensitivity reactors” or SPT negative. 

### 2.3. Relative Allergen Sensitivity

Patients were sorted based on their relative allergen sensitivity. The average number of identified allergens per person, gender, and age in each subgroup was compared in the table under results.

### 2.4. Immunotherapy

Patients with positive skin tests were offered AIT if their allergy symptoms were not adequately controlled by medications and avoidance, or if they wished to reduce medications. AIT was provided for all IDT positive allergens. 

The maximum possible concentration of each allergen, or the largest tolerated dose, was achieved within 4 months in all cases treated by subcutaneous (SCIT) or 2 months for those on sublingual immunotherapy (SLIT) [24,25]. Patients were re-evaluated every 6 months or sooner.

### 2.5. Patient-Reported Symptom Score

The primary outcome was the patient’s reported symptom score. Improvement among children was based on their parent’s report of symptom relief and diminution of episodes of recurrent OME. A 10-point forced-choice Likert-type questionnaire was administered pre-treatment and after achieving AIT maintenance. Patients were asked, “On a scale of 1 (minimal) to 10 (terrible) how would you rate your current symptoms?” This question meets all criteria for designing Likert scales [26]. We defined percentage improvement as (pretreatment symptom score minus symptom score after AIT)/pretreatment symptom score. 

### 2.6. Statistics

Statistical analyses were performed by a professional statistician using SAS 9.4 software (SAS^®^ Institute Inc., Cary, NC 27513, USA).

## 3. Results

One hundred ten patients met inclusion criteria for the study. Patient ages ranged from 3 to 70 years (59 females, mean 35.3 years, 51 males, mean 25.3 years). Fifty-three (48%) were children 3 to 15 years old (Table 1), of whom 72% were SPT negative (Table 2). The patient demographics of the two skin sensitivity groups (SPT+/IDT+ and SPT−/IDT+) were comparable except that there were far more (38 vs. 15) children in the low-sensitivity group (Table 1 and Table 2). 

### 3.1. Direct Comparison of SPT and IDT

There were 39 patients initially tested by certified allergists using only SPT and subsequently retested by us using IDT. Comparing tests for the same 12 allergens, SPT was found to detect only 16% of allergens found by IDT (Figure 1).

The two tests agreed as positive in 78 (20%) and negative in 88 (22%). IDT differed from SPT among 231 (58%) antigen tests as it detected 3.8 times more allergens (total of 334 vs. 55) per patient than SPT (9.5 vs. 2.5, *p* < 0.01). SPT exhibited a sensitivity of 16% of that of IDT, using IDT as the standard. Only 8% of initial SPT wheals were positive, and based on those tests AIT had been offered by the allergists to merely 8 (20.5%) patients. In contrast, IDT found all 39 to be allergic. AIT was recommended to all 39.

### 3.2. Allergy Testing Effectiveness of IDT

Skin testing in the 110 patients was positive for 833 of 1320 allergy tests placed. Forty-seven (43%) high-sensitivity (SPT+) patients were positive for only 81 allergens. IDT identified 305, or 3.8 times, additional treatable allergens among these high-sensitivity SPT+ patients (subgroups D&E) than did SPT alone (81) (Chi-Square *p* < 0.001, 95% CI: −0.47, −0.36), (odds ratio 0.09; 95% CI: 0.05 to 0.15) (Table 1, Figure 2). 

Identifying these allergens may have improved the chances of including all relevant allergens for AIT. IDT was significantly more sensitive than SPT (Chi-Square *p* < 0.001, 95% CI: −0.47, −0.36). 

Sixty-three low-sensitivity (SPT−) patients were positive only by IDT among low-sensitivity IDT+ (subgroups A, B, and C. Table 2, Figure 2), identifying an additional 447 potentially treatable allergens. These 63, 57% of all patients, would have been incorrectly identified as non-allergic by relying only on SPT. 

Children are especially likely to be missed if IDT is not done: 38 (72%) of the 53 children ages 3–15 expressed low sensitivity: SPT−/IDT+ (subgroups A, B, and C). Of these, 30 (83%) were in the lowest sensitivity subgroup A (Table 2). This demonstrates the necessity of testing children with the most concentrated dilution, 1:500 *w*/*v*, in order to diagnose allergy in young OME patients.

### 3.3. Response to AIT Is Independent of Skin Sensitivity

Pre-treatment and post-treatment symptom scores, and percent improvement for each group after reaching AIT maintenance, are recorded in Table 2. In the SPT+ high-sensitivity group (subgroups D, E) the average pre-treatment symptom scores were reduced from 8.13 to 2.75 (66% improvement). In the low-sensitivity IDT+ group (subgroups A, B, C), the average pre-treatment symptom scores were reduced from 8.51 to 2.95 (63% improvement). 

The difference proved to be not significant (Chi Sq *p* = 0.86 = NS; 95% Cl: −0.048, 0.040), demonstrating that the magnitude of AIT treatment responses based on IDT are the same as regardless of high- vs. low-sensitivity among patients. This held true when comparing high vs. low sensitivities among both the children (70% vs. 67%) and adults (60% vs. 59%). We found a significantly greater percent of children improved from AIT compared with adults, in both the high-sensitivity (70% vs. 60%) and the low-sensitivity (67% vs. 59%) groups (Table 2).

### 3.4. AIT Results Unaffected by Comorbid Allergic Diagnoses

We found 81% had two or more allergic diagnoses, including 79 (72%) with AR and 39 (35%) who had asthma (Table 3).

Of all 110 patients, only 21 (19%) had ETD as their sole allergic symptom and they reported the greatest improvement: 80%. ETD was the sole symptom among 58% of the 53 children. Chi-Square for difference *p* values comparing the Percent Improvement from AIT of patients with the same allergic comorbidities, grouped by relative allergen skin test sensitivity of the two groups (low-sensitivity (SPT−) and high-sensitivity (SPT+)), was not statistically different for any allergic comorbidity. Improvement was statistically not different for the total of study patients with OME/ETD (64.6%, *p* = 0.32 = NS), nor for those with comorbid allergic diagnoses AR = ETD (69%, *p* = 0.67 = NS) nor AR plus asthma + OME (63%, *p* = 0.98 = NS). 

### 3.5. Duration of AIT Benefit

Sixty-eight patients, previously reported, with OME/ETD or retracted tympanic membranes and who had either type B or C tympanograms [11], underwent a subgroup analysis. All 14 presenting with OME resolved as demonstrated by normal post-treatment otoscopy and tympanometry. Sixteen of 17 type B returned to type A, while all 19 type C returned to normal following AIT. Fifty-five patients completely and five partially resolved their ETD following AIT, maintaining resolution for four to seven years of follow-up long after their tubes had extruded. The eight failures were all ≥33 years old (average age 55.7). IDT test results for specific antigens for the 60 resolved patients are shown in Table 4. Of the 441 antigens detected among those who resolved, only 40 (9%) would have been detected by SPT.

## 4. Discussion

The purpose of this study was to demonstrate that the sensitivity of the skin testing method used to diagnose allergy is critically important in patients who have ETD symptoms.

Otitis media with effusion is the major form of chronic relapsing inflammatory disease of the middle ear. It is a disease of immense social and financial impact among families of young children, accounting for the most common surgical procedure requiring general anesthesia (ventilation tube insertion) for children [27]. Children with hearing loss secondary to OME constitute the largest group of people in the world with a reversible learning disorder. 

Current literature supports a strong association of allergy with ETD [4,7,14,28,29]. The most basic question that must be answered in each case of ETD requiring TTP is “What is the underlying pathophysiology?” We can surgically re-establish aeration with TTP, but curing the underlying condition requires knowledge of the pathophysiology. The middle ear space is an anatomic extension of the upper airway, and it is established that the middle ear is capable of mounting an inflammatory response identical to other areas of the respiratory tract. Mast cells and their mediator tryptase, both indicators of a Th2-driven allergic response, are present in a majority of ears that have chronic effusion [30] and, just as in the sinuses, the OME middle ear also has degranulating eosinophils [8].

The 2018 International Consensus Statement on Allergy and Rhinology [31] cites studies showing an association of allergy with both ETD and OME, stating, “The frequently observed clinical association of Eustachian tube symptoms and AR is corroborated by high-level evidence that demonstrates that in AR patients, nasal challenge with histamine or relevant aeroallergens results in transient Eustachian tube obstruction … This body of evidence supports a direct causal role for AR in some cases of Eustachian tube dysfunction … The middle ear mucosa may behave in a manner similar to nasal mucosa and be a site of local IgE-mediated inflammatory reactions.” Others conclude that OME is the result of allergic inflammation in the middle ear that creates mucosal and Eustachian tube edema [7,29].

The 2018 Consensus on Otitis Media [13] also showed an association of AR to OME but concluded, citing only two articles dealing with treatment limited to antihistamines, that “there is also no convincing evidence that directly treating allergy affects OME outcome.” The Consensus did not cite any of the 21 articles in reference [32] (Table 1) that recorded 2526 OME patients, 1553 tested by SPT or in vitro, and 773 tested by IDT, all with subsequent AIT, with respective resolution of about 60% in both groups. 

In order to resolve a patient’s OME, it is essential to identify that the patient is allergic. Clearly no matter how they were tested, once they were identified as allergic, immunotherapy was effective in resolving their OMD/EDT in more than half of patients reported from 1942 to 2008.

Studies that find no increased allergy in subjects with OME often rely on less objective criteria, such as history or patient questionnaires, than actual skin testing to arrive at a diagnosis of allergy. Tomonaga criticized many of these methodologically flawed studies. He found 21% of his kindergarden and elementary school patients had OME, of whom 87% were atopic by skin testing [33]. 

Although a majority of our ETD patients improved with AIT, we found that adults are less likely to improve than children. Anatomy of the pediatric Eustachian tube is different from that of adults, and their allergic inflammation may be less intense and/or of shorter duration. These facts are possible reasons to explain the better treatment responses in children (see Section 3.2 above).

The difference in expected outcome for adults that we report here in Table 2 had been demonstrated among the previously reported 68 patients. Among the children ages 4–15, complete resolution was experienced by 92%, whereas only 83% of those 16–50 and only 46% of those 51–70 experienced complete resolution after AIT [11] (Table 3). 

It is conjecture as to how much improvement would have been experienced among those high-sensitivity reactors (SPT+/IDT+) in Subgroups D and E if only the SPT+ allergens had been treated. These patients had 81 allergens positive by SPT but also had about four times more (305) allergens positive by IDT (Figure 2). If the treated SPT+ allergens included each patient’s critical allergens, then good results would likely have resulted, but the odds favor treatment with the greater number of allergens.

### 4.1. Identical Improvement with Treatment Based on SPT vs. IDT

The 2008 Practice Parameters [19] (Pg. S6) state, “Intracutaneous tests will identify a larger number of patients with lower skin test sensitivity and are used when increased sensitivity is the main goal of testing.” 

We found this to be true in that patients in Subgroup A who responded only to concentrated antigen (1:500 *w*/*v*) reported 66% improvement, suggesting that those positive responses were not false positives. Even when comparing the two extremes of skin sensitivity (Subgroups A vs. E, Table 2), there was still no demonstrable difference in degree of symptom improvement among either the children (Subgroup A 67% vs. Subgroup E 67%) or adults (Subgroup A 65% vs. Subgroup E 62%).

### 4.2. Arguments for Adding IDT to SPT

The need for IDT in the face of a negative SPT has long been debated [34] for at least four reasons. First, it is argued that ID tests for allergens not detected by SPT offer no further clinically relevant information [1]. Calabria’s detailed review stated, “For lower potency or non-standardized allergens, the ID skin test may identify a higher percentage of patients with lower levels of clinical sensitivity, and a positive test result may be more clinically relevant” [35]. 

Second, the relevant measure by which allergy tests should be judged is the patient’s clinical response to AIT based on their test results [36,37,38]. 

Third, AR patients who are SPT-negative have been diagnosed with non-allergic rhinitis with eosinophilia syndrome (NARES) rather than low-sensitivity allergy [39]. Yet, up to 60% of SPT negative NARES patients have been reported to respond to AIT based on IDT [40].

Fourth, Kaffenberger acknowledged that it took SPT-based AIT patients an average of 265 days longer to reach maintenance [41], or 38 additional weekly visits. Thus, the long-term total of testing plus treatment cost difference favors IDT.

### 4.3. Study Strengths

There are six major strengths of this study. First, there has been no prior large study that has compared the effectiveness of adding IDT to SPT using treatment outcomes based on the test results.

Second, we found a robust benefit from AIT based on IDT detection of allergens, averaging 64% symptom improvement (Table 2 and Table 3).

Third, using IDT, rather than less sensitive tests, doubled the overall number of patients and more than tripled the number of children diagnosed as being allergic (Table 2 and Table 3). 

Fourth, the failure rate of 5.7% among low-sensitivity reactors (SPT−/IDT+) was statistically identical to the 5.1% experienced by the 47 highly sensitive SPT+ patients. (Chi Sq *p* = 0.51; 95% CI: (−0.032, 0.064)). This indicates that IDT results were not false positives.

Fifth, 14 patients in this study served as their own control. Although they had become free of effusion, otalgia, or drainage during AIT, their symptoms or abnormal tympanograms recurred when they stopped their AIT prematurely. All again resolved after resuming their AIT. 

Finally, a second control group of 21 previously reported OME patients [11] who declined AIT all failed to resolve their otalgia, conductive hearing loss, or OME. 

### 4.4. Study Limitations

There were several study limitations. First, the study was neither randomized nor blinded, so has risk of bias, and the two control groups were self-selected.

Second, we chose to use a simple 10-point Likert scale to measure AIT outcomes. A Likert scale was chosen because of simplicity, validity, and especially ease of clinical use [26,42], although QOL scores tend to have more statistical power [43,44,45]. Unlike our study, Kaffenberger measured AIT treatment outcomes following IDT using a multi-question quality of life (QOL) instrument, but only 14 of their 37 patients who reached maintenance completed the full questionnaire, resulting in a small study with insufficient power for definite conclusions [41].

Third, although one third (39/110) of our patients had been skin tested by both methods, the absence of actual SPT testing due to the procedures of the specific practice studied did not allow for a direct comparison between SPT and IDT sensitivity among the other 71 patients. 

Fourth, none of the patients had ET endoscopy. However, of the 9 adult failures, average age 53, 5 were sent for ET evaluation and none were found to be candidates for ET dilatation.

### 4.5. Summary

The data from these 110 patients with OME/ETD strongly supports our hypothesis: IDT demonstrated a markedly greater sensitivity for detecting allergens that are clinically relevant for AIT. Furthermore, AIT treatment of low-sensitivity patients (Subgroups A, B, and C), whose allergies could only be detected with IDT, reported nearly identical symptom improvement as compared to treating high-sensitivity patients (Subgroups D, E) who did not require higher concentration IDT tests for diagnosis. 

## 5. Conclusion

Use of only SPT in our patients would have failed to identify 57% of ETD individuals as having allergic disease, including 72% of children ages 3–15 who were SPT negative. Adding IDT following negative SPT doubled the number of patients diagnosed as being allergic (Figure 2) and almost tripled the number of children (Table 2). Use of only SPT would also have missed 91% of IDT positive antigens that might have been critical to successful AIT treatment (Table 1 and Table 2) (Figure 2). 

Patients diagnosed by IDT reported 63% symptomatic improvement for their ear symptoms from AIT, which was statistically not different from the 66% improvement reported by patients diagnosed by SPT (*p* = 0.86 = NS). Reliance solely on SPT may help explain why many otologists do not consider allergy as a possible treatable etiology for their patient’s OME/ETD. These data strongly support increased utilization of intradermal testing and invite additional clinical outcome studies.

## Figures and Tables

**Figure 1 diagnostics-11-00763-f001:**
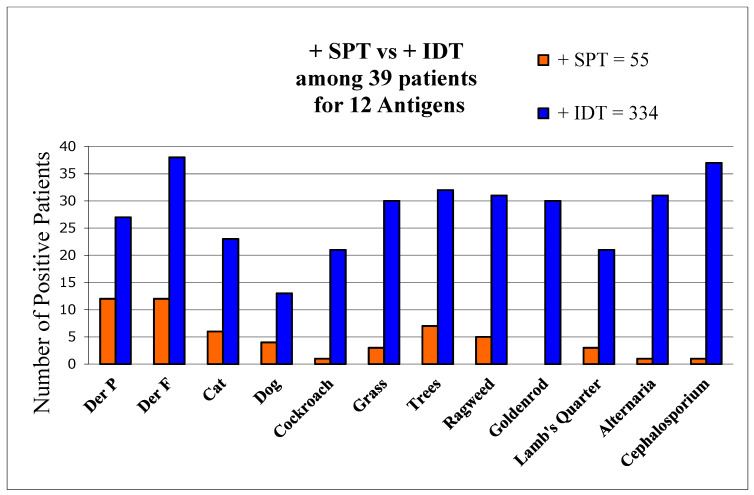
Comparisons of allergen detection by both SPT and IDT. Number of positive skin-test reactions by SPT and IDT to each of 12 allergens among 39 patients tested by both methods.

**Figure 2 diagnostics-11-00763-f002:**
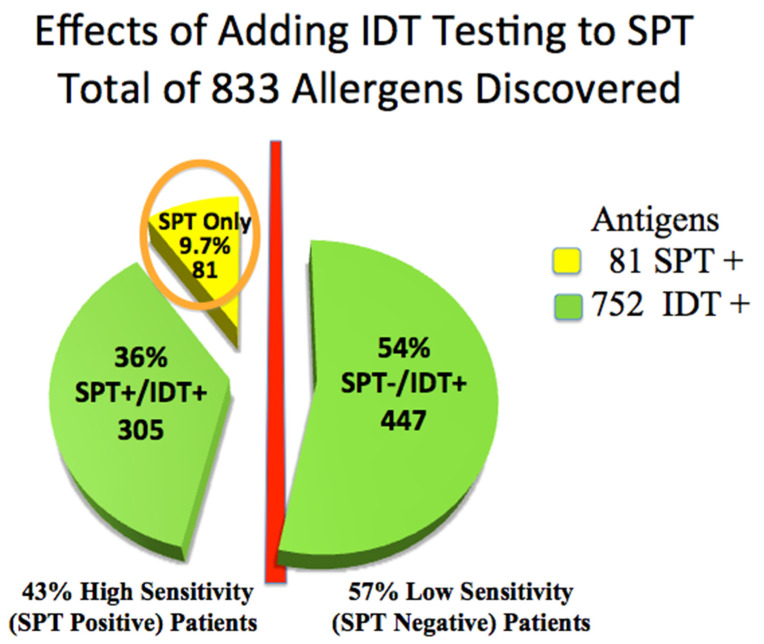
Advantage of adding IDT testing to SPT: total allergens discovered. Circled: 81 antigens positive by SPT.

**Table 1 diagnostics-11-00763-t001:** Effect of adding IDT testing in SPT+ and SPT− patients.

		Low-Sensitivity Subgroups A, B, C	High-Sensitivity Subgroups D and E
Patients (%)		63 (57)	47 (43)
Total # Antigens	IDT+	447 (54)	305 (34.5)
	SPT+	0	81 (9.7)
Avg. IDT+ Antigens *		7.1	6.48
Male # (%)		32 (51)	19 (40)
Female # (%)		31 (49)	28 (60)
Avg. Age in Years		30.2	31.3
Patients 3–15 Years Old		38 (60)	15 (32)
Patients 16–50 Years Old		16 (25)	19 (40)
Patients 51–75 Years Old		9 (15)	13 (28)

* The average number of antigens detected by IDT was significantly greater than those detected by SPT alone in both sensitivity groups. (*t* test *p* = 0.004, 95% CI: (−2.54, −0.506). This detection advantage of IDT was even greater for the high-sensitivity (SPT+) group vs. the low-sensitivity (SPT−) group.

**Table 2 diagnostics-11-00763-t002:** Percent Improvement in Ear Symptoms at AIT Maintenance by Allergen Sensitivity Subgroup and Age.

	Low-Sensitivity (IDT+/SPT−)				High-Sensitivity (IDT+/SPT+)		
Subgroup	A	B	C	Total	D	E	Total
Strongest+	Only D2	D2+ 1 D3	D2, D3	SPT−	D2, D3, +1 D4	D4	SPT+
**CHILDREN**							
Age 3–15	22	8	6	36 (71)	7	8	15 (29)
SYMPTOM SCORE							
Before AIT	8.7	8.3	9.5	8.3	7.5	8.9	8.2
After AIT	2.8	1.9	2.8	2.5	2.4	3.0	2.7
**% Improvement**	**67%**	**74%**	**71%**	**70%**	**67%**	**67%**	**67%**
**Adults**							
Age 16–70	6	8	13	27 (46)	11	21	32 (54)
SYMPTOM SCORE							
Before AIT	8.5	9.2	8.3	8.6	7.9	8.5	8.0
After AIT	3	3.4	3.6	3.5	3.6	3.1	3.3
**% Improvement**	65%	62%	57%	**60%**	53%	62%	**59%**
**Total PATIENTS**	**28 (45)**	**16 (25)**	**19 (30)**	**63 (57)**	**18 (39)**	**29 (61)**	**47 (43)**
SYMPTOM SCORE							
Before AIT	8.2	8.19	8.6	8.51	7.75	8.5	8.13
After AIT	3.3	2.66	3.3	2.95	3.2	2.1	2.75
**% Improvement**	**66%**	**66%**	**62%**	**63%**	**58%**	**72%**	**66%**

(Percent) Skin Test Responses: A = all D2 (1:500 *w*/*v*), B = D2 and 1 D3 (1:2500 *w*/*v*), C = D2 and D3, D = 1 D4 (1:12,500 *w*/*v*), and E = 2 or more D4.

**Table 3 diagnostics-11-00763-t003:** Percent improvement from AIT of patients with the same allergic comorbidities grouped by relative allergen skin test sensitivity.

	Low-Sensitivity (Subgroups A, B, C)			High-Sensitivity (Subgroups D and E)			Total Patients		
COMORBIDITY	No of Patients	Average Age	% Improved	No of Patients	Average Age	% Improved	No. of Patients	% Improved	*p* Value
ETD Alone	16	22.4	78%	5	21	83%	21	80%	ID
AR + ETD	28	35.5	63%	22	32.6	67%	50	69%	*p* = 0.67 = NS
Asthma + ETD	5	32.7	78%	5	37.5	68%	10	73%	ID
AR + Asthma + ETD	14	28.6	62%	15	30.9	63%	29	63%	*p* = 0.98 = NS
Total	63	30.2	61.8%	47	31.3	66.7%	110	64.6%	*p* = 0.32 = NS

NS = not significant. AR = allergic rhinitis, ETD = eustachian tube dysfunction, ID = insufficient data.

**Table 4 diagnostics-11-00763-t004:** Effect of adding IDT testing in 60 OME patients who resolved with immunotherapy [11].

	SPT			% + SPT	DT		
CLASS + IDT	D6	D5	D4		D3	D2	TOTAL
DER P		2	6	1.8		32	40
DER F	2	1	9	2.7		47	59
CAT					3	26	29
DOG					2	15	17
COCKROACH	1	1	1	0.7	7	29	39
GRASS		2	3	1.1	3	30	38
TREES			2	0.4	6	25	33
RAGWEED	2	1	4	1.3	3	31	41
GOLDENROD		1			7	28	36
LAMB’S QUARTER			1	0.2	3	17	21
ALTERNARIA					3	31	34
CEPHALOSPORIUM			1	0.2	15	38	54
# Allergens Found	5	8	27	40	52	349	441
% Of 441 Total	1.1%	1.8%	6.1%	9%	11.8%	79.1%	100.0%

## Data Availability

The data presented in this study are available on request from the corresponding author, David Hurst. The data are not publicly available due to privacy of patient identity.

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
