# Peer review of "Intradermal Testing Doubles Identification of Allergy among 110 Immunotherapy-Responsive Patients with Eustachian Tube Dysfunction"

_diagnostics, 2021, doi:10.3390/diagnostics11050763_

Round 1

Reviewer 1 Report

The topic is interesting and worth investigation; the paper is well written and clear. Paper can be accepted in the present form

Reviewer 2 Report

The allergy is well known one of the factors of OME or ETD. And Dr. Hurst is pioneer in the role of allergy in the pathogenesis of OME. 

  1. It would be better if patients were divided child and adults. Because the basic situation of the pediatric Eustachian tube status is different from adult.
  2. In the introduction or discussion, diagnostic method for allergy using MAST or other test by blood sampling, and describe the difference from skin pinprick test or ITD. 
  3. In children, skin pin prick test is aggressive than MAST or FAST. 
  4. If  you perform ET function test or ET endoscopy for allergy tested patients, please add and correlation. If you did not simultaneously, please describe in the discussion. 
